# Edge Cooling of a Fuel Cell during Aerial Missions by Ambient Air

**DOI:** 10.3390/mi12111432

**Published:** 2021-11-21

**Authors:** Lev Zakhvatkin, Alex Schechter, Eilam Buri, Idit Avrahami

**Affiliations:** 1Department of Mechanical Engineering and Mechatronics, Ariel University, P.O. Box 3, Ariel 44837, Israel; phinistsokol@gmail.com (L.Z.); ghko.ghsu@gmail.com (E.B.); 2Department of Chemical Sciences, Ariel University, P.O. Box 3, Ariel 44837, Israel; salex@ariel.ac.il

**Keywords:** fuel cell, edge-cooling, aerial missions, CFD

## Abstract

During aerial missions of fuel-cell (FC) powered drones, the option of FC edge cooling may improve FC performance and durability. Here we describe an edge cooling approach for fixed-wing FC-powered drones by removing FC heat using the ambient air during flight. A set of experiments in a wind tunnel and numerical simulations were performed to examine the efficiency of FC edge cooling at various flight altitudes and cruise speeds. The experiments were used to validate the numerical model and prove the feasibility of the proposed method. The first simulation duplicated the geometry of the experimental setup and boundary conditions. The calculated temperatures of the stack were in good agreement with those of the experiments (within ±2 °C error). After validation, numerical models of a drone’s fuselage in ambient air with different radiator locations and at different flight speeds (10–30 m/s) and altitudes (up to 5 km) were examined. It was concluded that onboard FC edge cooling by ambient air may be applicable for velocities higher than 10 m/s. Despite the low pressure, density, and Cp of air at high altitudes, heat removal is significantly increased with altitude at all power and velocity conditions due to lower air temperature.

## 1. Introduction

Hydrogen-based fuel cells (FCs) are considered highly efficient, non-toxic, and non-polluting alternative energy conversion devices. In a combined fuel and fuel-cell system, their theoretical energy density may be in some cases 10 times higher than lithium-ion batteries, which makes them extremely attractive for portable devices, small vehicles, etc. [1]. Among all types of vehicles that can be powered by the FC power plant, lightweight unmanned aerial vehicles (UAVs) should be signified as those that could benefit the most from the energy density and capacity. UAVs conventionally use LiPo batteries as an energy source, which limit their normal flight duration to about 20–30 min for multirotor vertical take-off and landing drones and 60–90 min for fixed-wing drones [2,3].

It was shown [1,2,3,4,5] that a FC-based energy system has a higher energy density than LiPo batteries, which are considered to be the most efficient batteries on market (with up to 200 Wh/kg).

While generating electricity, most proton-exchange membrane fuel cells (PEMFC) inevitably emit a certain amount of heat in a non-uniform or non-stable manner. Thermal management of PEMFCs is a major issue for efficient performance and preventing components’ damage. Therefore, the issue of PEMFC colling may be critical [6], especially in specific applications [7].

Usually, cathode opened to the air (“air-breathing”) type PEMFCs are cooled by the oxygenated air that is flowing through the bipolar plates (BPPs). However, in some aerial missions, the ambient air might damage the FC membranes and thus, the FC should be enclosed in a protective dome with a controlled ventilation intake for oxygen supply. This approach keeps the FC safe from a dusty environment, prevents dehydration of the membranes due to extremely dry or cold ambient air, and may be used when recycling the exhausts of water vapor required for the production of hydrogen by hydrolysis [8,9]. In such cases, the oxygenated air passing through the FC BPPs is isolated from the environmental air, and therefore FC cooling becomes a critical issue with removing the heat from the stack’s edges. Various methods are applied for cooling the FC, such as passive methods, air cooling, liquid cooling, and phase-change cooling [7].

An edge cooling method by convection over FC stack was suggested by several studies [10,11,12]. Indeed, it is not suitable for the majority of applications since it increases the volume of the stack and requires a high flow of air. Yet, it was proven to be effective for high-altitude fixed-wing UAVs, where the ambient flow may be utilized for heat removal. Such an application was suggested by Barroso et al. [13], who utilized a closed type of high-temperature stack with metal BPPs. The stack was installed inside the drone’s fuselage and was cooled down by ambient air that was supplied to it from the air intake.

Several studies suggested modifications to the FC with graphite BPPs [14], which have higher thermal conductivity (from 20 to 400 W/m∙K). In these studies, BPPs themselves had fins and FCs had a maximal surface area of edges to increase the heat transfer rate. However, this approach requires a special design of the FC and does not apply for commercial off-the-shelf FCs. Moreover, it cannot be scalable for smaller scales (e.g., small drones).

When off-the-shelf FCs are used, external radiators can be attached to the FC edge to increase the heat exchange and improve thermal management. The radiator’s geometry and accordingly weight depend on specified requirements such as minimal drone speed, maximal ambient temperature, FC power output, and desired working temperature range. For standard commercially available stacks, the normal temperature range is usually between 45 and 60 °C. Overcooling will result in lower performance, while overheating may cause membrane damage or catalyst deactivation.

In this paper, we describe an edge cooling system for fixed-wing UAVs derived by air-breathing PEMFC. The system utilizes ambient air to remove FC heat during flight. This method can be potentially scalable and fit a large variety of commercial PEMFCs. In this study, we examine the applicability and limitations of a radiator cooling method using a numerical model validated by experiments.

## 2. Materials and Methods

To prove the feasibility of the suggested edge cooling concept, a combined approach of numerical and experimental analyses was used to model the airflow over a commercial PEMFC. Since the experiments were limited to a wind tunnel at ground air conditions (Figure 1), in the first stage we developed a computational fluid dynamic (CFD) simulation the flow inside the experimental wind-tunnel channel and over the FC for validation with the experiments. At the next stages, more realistic CFD models of the drone in flood flow were used to estimate heat regime at elevated altitudes and for design optimization.

### 2.1. Wind Tunnel Experiment

To test the applicability of the edge cooling method for a fixed-wing UAV, a set of experiments in the aerodynamic tunnel was performed using a graphite BPP 200 W PEMFC (BCH energy, Jiang-su, China). The results of these experiments were compared with a CFD model of the same dimensions for validation of the numerical method.

#### 2.1.1. Experimental Set-Up

An experimental system was built to model edge cooling over an FC with radiators attached to its top and bottom edges (Figure 2a). To illustrate the edge cooling approach by convection to high-velocity ambient air, a model of an FC inside a wind tunnel was developed.

An air compressor (PVL 352 T, DYNAIR Industrial Ventilation, Maico, Italy; Figure 2e) was used with a flow rate capacity of up to 30 m^3^/min, to which a rectangle channel was attached, providing the airflow of up to 20 m/s. The tunnel was built from 0.5 cm thick polymer foam panels covered with smooth paper (Figure 2d) and had the following dimensions: nozzle length was 50 cm; the tunnel length was 150 cm, and the tunnel cross-sectional dimensions are 16 cm × 8.25 cm. The FCs cross-sectional dimensions were 4 cm × 8.5 cm, which leaves a 4.5 cm distance between the tunnel and the FC surfaces.

The FC was placed inside a 3D-printed cowl that simulated a drone’s fuselage (Figure 2c). The airflow inside the tunnel was stabilized by a grid installed after the nozzle with a cell size of 4 mm. A dynamic anemometer (AVM-03, Prova Instruments Inc., New Taipei City, Taiwan) placed 20 cm before the cowl was used to measure air velocity in the tunnel. Ambient air temperatures were in the range of 30–32 °C.

Aluminum heat sink radiators (40 mm × 40 mm × 11 mm, Shenzhen Gdstime Technology Co., Ltd., Guangdong, China; Figure 2b) were glued together three in a row to cover each of the two edges of the FC stack (Figure 2a). To assure efficient heat transfer, the radiators were attached to the stack by Heatsink Plaster (STARS-922, LabsGuru Technologies Pvt. Ltd., Uttar Pradesh, India), with a thermal conductivity of 1.2 W/m∙K, which also electrically insulated between the stack’s surface and the radiators. The FC was provided with a hydrogen supply from a compressed H_2_ tank at flow rates 0.5–1.5 L/min and 0.5 bar pressure. The FC power output was controlled by a programmable DC load (model 8502, B&K precision Corp., Yorba Linda, CA, USA) for working at output power conditions of 0–110 W. To allow sufficient oxygen supply to the cell, a DC air pump (Guangdong KYK Technology Co., Ltd., China) was attached to the dome’s sidewall to suck ventilation air through the BPPs at a flow rate of up to 8 L/min (Figure 2c).

BPP temperatures were measured using two thermocouples (k-type), placed at the FC center and below the radiator (Figure 2a). In addition, inlet and outlet air ventilation temperatures (pumped over the FC) were measured (Figure 2c,d). Twelve experiments were performed with different FC heat outputs (60 W, 90 W and 125 W) and at various airflow velocities (5–20 m/s) inside the tunnel. At each experiment, it took up to 5 min for the FC temperatures to stabilize.

#### 2.1.2. Estimation of Heat Generated and Removed

Assuming that water products exit the fuel cell’s cathode in their vapor form and that all of the free energy of the H_2_ reaction was converted into electrical energy, the Nernst Equation (1) under standard conditions [15] can be expressed as:(1)$$E=\frac{-\Delta {\bar{h}}_{f}}{2F}$$
where $\Delta {\bar{h}}_{f}$ = −241.83 kJ/mol is the *calorific value* or change in enthalpy of formation for burning of hydrogen and forming steam (using “*lower heating value*”, LHV), *F* is Faraday constant (96,485 C/mol). The theoretical output voltage of each cell would be 1.25 V under standard conditions [15].

The difference between this theoretical voltage and the actual cell voltage represents the energy that is not converted into electricity—that is, the energy that is converted into heat instead. Therefore, the heat generated by the FC stack can be estimated using Equation (2) [15]:(2)$$Q_{FC}=NI\left(1.25-\frac{V_{FC}}{N}\right)\left[W\right]$$
where *N* is the number of the membrane electrode assembly (MEA) electrochemical cells in the FC, *I* is current, and *V_FC_* is FCs voltage.

### 2.2. Numerical Model of the Experiments

A numerical model of the airflow inside the experimental wind-tunnel channel and over the FC was developed for validation with the experiments. The CFD finite-volume numerical method was employed using Ansys Fluent^®^ software (2021 R2, ANSYS, Inc., Canonsburg, PA, USA) to solve the flow and heat regime in a model of the FC inside the tunnel. A steady-state and incompressible flow was assumed. SST k-ω turbulence model [16] was chosen since it is proven to be well suited for solving problems of radiator cooling with an external flow [17,18]. The turbulence model was utilized with default solver settings with the governing continuity, momentum, and energy Equations (3):(3)$$\begin{array}{c} \nabla \cdot U=0\\ \rho \left(\frac{\partial U}{\partial t}+U\cdot \nabla U\right)=-\nabla P+\mu \cdot \nabla^{2}U\\ \rho C_{p}\left[\frac{\partial T}{\partial t}+\left(U\cdot \nabla \right)T\right]-k\nabla^{2}T=q \end{array}$$
where *P* is the static pressure, **U** is the velocity vector, *T* is temperature, *t* is time, *q* is heat flux, *μ*, *ρ*, *Cp* and *k* are viscosity, density, specific heat capacity, and thermal conductivity, respectively.

Assuming symmetry, the geometric model represented one-half of the FC and tunnel actual setup, including three domains (Figure 3): airflow domain (blue), FC domain (grey), and radiators (orange). The radiator was modeled with the dimensions of radiators used in the experiments (Figure 2b). The air domain cross-sectional dimensions were the same as those of the tunnel used in experiments and its length was 32.5 cm, leaving an 8.5 cm distance between the inlet and the aerodynamic cowl.

Boundary conditions included pressure outlet, velocity inlet (according to the velocities measured in the experiments), symmetry, and wall conditions, as presented in Figure 3a. The FC domain was set as a constant heat source with *Q_FC_* as calculated from Equation (2) for each case. The air temperature was set to 30 °C.

Meshing was performed using an unstructured tetrahedral grid with inflation layers on all the surfaces in touch with airflow. The inflation contained 18 layers with a 1.2 growth ratio. The first layer height was 0.018 mm (based on Y+ calculation for air velocity of 20 m/s [19] (p. 467)). After examining several mesh configurations (as detailed in Appendix A), the chosen mesh included 1,846,671 finite volumes with an average size of 7.5 mm and is presented in Figure 3b,c.

The thermal conductivity of the aluminum radiator was set to 202.4 W/m∙K; the thermal conductivity of graphite was 400 W/m∙K. The 2 mm thermal grease layer thermal conductivity between the FC and the radiators was set to 1.2 W/m∙K.

### 2.3. Onboard Cooling with a Modified Radiator

To evaluate the performance of the FC edge cooling approach at realistic onboard conditions, a numerical model of a fixed-wing drone fuselage was built in an ambient air domain. The same numerical method was applied for the geometry shown in Figure 4c of a drone’s fuselage with the radiator connected to its surface. The total length of the designed fuselage is 85.7 cm, and it is symmetrical with respect to YZ and ZX planes to allow a quarter-domain simulation.

This model included modifications to the radiator’s geometry (Figure 4a,b). The fins were elongated from 11 mm to 15 mm (from the base) to allow better heat transfer, and the distance between neighboring fins was increased from 2.25 mm to 4.54 mm to reduce the radiator’s resistance to ambient flow and increase air velocities between the fins. In addition, unlike the experiment, here the FC (and the attached radiators) were oriented perpendicular to the flow direction.

The geometry for simulation is presented in Figure 4d. The FC domain was modeled here using a constant working temperature boundary condition of 50 °C, attached to the radiators’ flat surface. Y^+^ for the inflation layer was 0.012 mm, assuming a maximal velocity of 40 m/s. An example of the mesh with a magnification view near the fins is presented in Figure 4e. The air domain includes two symmetry boundaries, and a no-slip boundary condition was applied to the fuselage wall. Velocity inlet and pressure outlet were set accordingly ahead and behind from fuselage, and the other two boundaries were set as zero-shear stress. Four different radiator’s locations were examined, located at a distance of X_radiator_ = 45.5 cm, 61 cm, and 76.5 cm from the fuselage front edge (Figure 4c). Four different inlet velocities were examined (10 m/s, 20 m/s, 30 m/s and 40 m/s).

In the CFD analysis, velocity and temperature distributions were examined, and drag force due to the radiator and the total heat removed were calculated and compared. The total drag force of each case (*D*) was obtained by integration of pressure distributions in the flow field, multiplied by four to account for the full geometry (out of the quarter model). The drag coefficient for each case was obtained using the Equation (4):(4)$$C_{D}=\frac{D}{\frac{1}{2}\rho V^{2}A_{D}}$$
where *ρ* is the air density, *A_D_* is the fuselage frontal area, and *V* is the cruise speed.

The total heat transfer (*Q*) was calculated using an Equation (5) by integration of the heat flux (*q*) over the area of the FC edge in contact with the radiators (*S_FC_*), and it was multiplied by four to account for the full geometry.
(5)$$Q=4\times \int_{S_{FC}}q\cdot dA$$

Convective heat transfer coefficients *h_c_* were calculated for each case using the Equation (6):(6)$$h_{c}=\frac{Q}{S_{FC}*\left(T_{FC}-T_{0}\right)}$$
where FC temperature is *T_FC_ =* 50 °C and ambient air temperature is *T*_0_
*=* 30 °C.

### 2.4. Heat Removal as a Function of Altitude

To study the effect of a drone’s altitude on heat transfer, the air properties were changed according to the standard Earth atmospheric model [20], which defines the relation between atmospheric temperature and pressure to altitude, Equations (7) and (8)
(7)T=15.04−0.0649h
(8)$$p=101.29\cdot {\left[\frac{T+273.1}{288.08}\right]}^{5.256}$$
where *p* is pressure (in kPA), *T* is the temperature in degrees Celsius, and *h* is the altitude in meters. The pressure and temperature values calculated for altitudes 0–5 km are presented in Table 1. These calculated values of pressure and temperature were utilized for parametric study in the numerical model of radiator located in *X*_radiator_ = 45.5 cm. Air density and heat capacity were calculated automatically by the program for each case using the kinetic gas theory (also detailed in Table 1). The total heat removed at each case was calculated using Equation (5).

## 3. Results

### 3.1. FC Performance: Power and Heat

To estimate the FC performance and total heat output, the voltage and output power were measured, total heat output was calculated (using Equation (2)), and these are presented in Figure 5 for 12 different current values (0–6 A). The voltage reduced with the current (from 32 V to 20 V), output power increased up to 110 W, and the calculated heat output reached up to 140 W.

### 3.2. Validation of the CFD Model Using the Experimental Results

Figure 6 shows a comparison between the experimental measurements and the simulation results for FC maximal temperature at different air velocities (5–20 m/s) and different FC power (60, 90, 125 W). FC maximal temperatures varied between 39 °C (for air velocity of 20 m/s and FC heat of 60 W) to 67 °C (for an air velocity of 20 m/s and a FC heat output of 60 W). In all the cases, the FC temperature reduced with air velocity and increased with consumed power. Note that both in the CFD simulations and in the experiments, the observed FC temperature distribution was relatively uniform, within the range of ±2 °C. These temperature distributions are manifested by the error bars in the graph.

The average difference between the numerical and the experimental temperature results is about 5 °C, which might be a result of variations in ambient air temperature, inaccuracies in fin dimensions, and possibly some hydrogen crossover (<10%) through tiny holes in the membrane and reaction with oxygen on the cathode side, and thus increasing the FC heat output. Other sources may include some rotational flow in the wind tunnel (despite the stabilization grid), some radiation heat transfer from the radiator, and some contact resistance of the connecting paste between the radiator and the FC non-uniform edge.

Figure 7 shows an example of the temperature and velocity distribution obtained from the numerical simulations for the case of 20 m/s inlet velocity and 125 W FC heat output, on the side surface of the FC (Figure 7a) and air domain (Figure 7b) and at a surface 1 mm away from the FC (marked using dashed line in Figure 7a). Thanks to the high thermal conductivity of the FC, the calculated temperatures of the FC edge were within a small range of 46–47 °C with a peak value at the FC center (marked in Figure 7a). Some insignificant effects of the thermal grease layer can be observed between the FC and the radiator.

However, the air temperature in contact with the radiator fins reduced only to 42–33 °C. This may be due to the low velocities (0–26 m/s) of airflow through the radiator fins (Figure 7b,d). The average velocity slightly increased with increased distance from the base of the radiators and may reach 40 m/s at the upstream edge. However, the general flow velocities between radiator fins along the flow were almost zero and the freestream flow produced a kind of a border layer from both sides of the radiator. Moreover, the temperature difference over the radiator’s fins was less than 2 °C. These results imply that for better heat transfer, the radiator fins should be extended in height and the spaces between them should be enlarged.

### 3.3. Onboard Cooling

The results for the onboard model of the fuselage in the ambient air domain are presented in Figure 8 for 20 m/s inlet velocity. Velocity and temperature distribution over the radiators (5 mm above fuselage surface) are shown in Figure 8b. In addition, in this model some reduction of downstream velocities was found by the radiators, with the fins’ rows forming a boundary layer above them. However, the average flow velocities inside the radiator were significantly increased (up to 25 m/s) compared to the wind tunnel experiment, resulting in better temperature distribution near the fins (down to 30 °C). This implies that the modifications of the radiators’ geometry are sufficient.

The effects of the radiator’s location along the fuselage and the cruise speed on drag force and total heat removed are shown in Figure 9. As the speed increased, both the heat removed and the drag force increased. The radiator improved heat removal dramatically but did not change the drag force (only when located at a far downstream location) in respect to the case with no radiators. The obtained drag coefficients, as calculated using Equation (4) ranged from 0.318 (for X = 45 cm at 40 m/s) to 0.453 (for X = 76.5 cm at 10 m/s).

The obtained values of total heat removal ranged between 210 W (for a radiator located at the nearest location, X = 45.5 cm at a cruise speed of 40 m/s) to 14.8 W (with no radiator at a cruise speed of 10 m/s) for 30 °C ambient temperature.

The obtained total heat transfer results for 10, 20 and 30 m/s drone speeds at altitudes of 0–5 km are presented in Figure 10. Although the air density and Cp decreased with altitude, in all the cases the total heat removed increased with altitude due to ambient temperature decrease.

For drone speed of 10 m/s, the heat removed changed from 150 W on the ground to 180 W at 5 km altitude. For 20 m/s, it increased from 245 W on the ground to 300 W at 5 km, and for 30 m/s it increased from 320 W to 400 W in 5 km altitude. These results imply that the edge cooling approach at typical cruise speeds improves at high altitudes.

Note also the differences for the cases on the ground with 30 °C (Figure 9) and with 15 °C (Figure 10).

Figure 11 shows a results summary for the total heat transfer and heat transfer coefficient (from Equations (5) and (6)) for the cases without a radiator (at 30 °C on the ground), with a radiator on the ground (at 30 °C and 15 °C), and at high altitude (5 km at −17.5 °C and 49.5 kPa) at different cruise speeds (10, 20 and 30 m/s). It is clearly shown that the radiator significantly improves the cooling efficiency for all the flight regimes. For the cases with different ambient temperatures and pressure, the heat transfer coefficient is reduced at high altitudes (Figure 11b), but the total heat transfer increases with the low temperatures due to higher temperature differences (Figure 11a).

## 4. Discussion

In this study, we suggest an edge cooling approach for removing heat from the FC stack in PEMFC-driven fixed-wing drones. Heat is removed by radiators placed on the FC edge and exposed to ambient air at cruise velocity. In a combined experimental and numerical analysis, we showed that the concept of removing FC heat by convection over radiators is feasible for a drone’s velocities higher than 10 m/s and ambient temperatures lower than 30 °C. Moreover, we showed that when the drone is flying at high altitudes, the heat transfer is improved due to the low air temperature.

The study has some limitations. The obtained heat removed (Equation (5)) does not take into account heat removed by the ventilated air (through the BPPs and driven by the pump). From calculations based on the temperature difference between intake and outtake and the mass flow rate, the total heat transfer through the FC was negligible (~1 watt) and thus can be omitted.

In addition, some hydrogen (<10%) might have leaked through tiny holes in the membrane and reacted with oxygen, thus increasing the FC heat output. Other limitations of the experiments may be due to some rotational flow in the wind tunnel (despite the stabilization grid), some radiation heat transfer from the radiator, and some contact resistance of the connecting paste between the radiator and the FC non-uniform edge.

The numerical models may have some limitations as well. The assumption of steady-state flow and symmetrical boundary conditions may not be accurate (due to vortex shedding from the radiator fins) and the assumption of constant wall temperature in contact with the radiator (in the onboard model), and the turbulent model may add some error in the calculations.

However, these limitations should not affect the overall conclusions of the study. From the numerical analyses, it was concluded that when the FC and radiators are located at the drone’s front, the obtained performance (according to drag force and heat transfer) is slightly better in respect to the central and rear locations. In addition, it was shown that radiators with double-space fins have better performance in heat removal than the standard ones.

The suggested method may be useful in cases when a commercial PEMFC is used, and the FC should be isolated from the ambient air to prevent FC damage or dehydration by dusty or extremely dry ambient air. FCs’ membrane humidity level is a crucial variable to improve the performance and to avoid damage to the membranes of PEMFC. The proton conductivity is directly proportional to the water content inside the membrane. Thus, there must be sufficient water content in the polymer electrolyte membrane. In general, PEMFC with extra humidification works 20–40% more efficiently. Hence, applying the proposed method is assumed to improve the water management inside the FCs and therefore its performance. Moreover, the exhausts’ water vapor can be collected and recycled for the production of hydrogen by hydrolysis and thus improve the energy density of the hydrogen generator [21]. In addition, it has the potential for scalability for drones of larger or smaller sizes [8,9]. Other non-FC-related applications for such a cooling system may be any heat generating device installed on a fixed-wing drone (e.g., battery, engine, CPU, etc.).

Future research will examine the further improvement of heat transfer by placing the drone propellor in the front of the drone, and a temperature control system should be developed to avoid over-cooling that might lead to FC operation stop.

## 5. Main Conclusions


The method may be applicable for on-board edge cooling for velocities higher than 10 m/s;The radiator significantly improves heat transfer from the FC to the ambient air;Despite the low-pressure density and Cp at high altitudes, heat removal at high altitudes is significantly higher at all examined power and velocity conditions due to lower air temperature;Location of the FC and radiators at the drone’s front has some benefit over the central and rear locations;The drone’s drag force is not affected by the radiator;Radiators with double-space fins exhibit better performance in heat removal.


## Figures and Tables

**Figure 1 micromachines-12-01432-f001:**
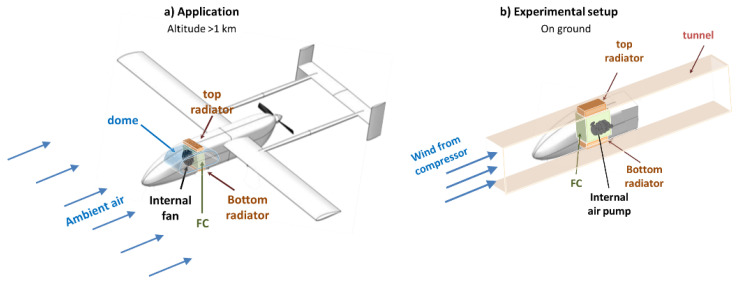
A schematic description of the application. (**a**) FC edge cooling by ambient airflow, using radiators, and of the experimental setup; (**b**) The FC placed in a phantom of a drone’s nose inside a wind tunnel.

**Figure 2 micromachines-12-01432-f002:**
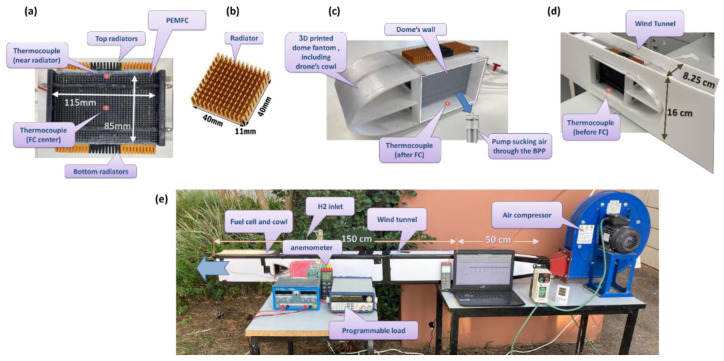
The experimental setup. (**a**) The PEMFC stack with radiators connected to the edges. Dimensions and thermocouple’s locations are shown; (**b**) Radiator; (**c**) The 3D-printed cowl with FC inside, post-FC thermocouple and pump’s locations are shown; (**d**) Wind tunnel walls (top wall not shown in the photo) with the cowl and FC location of pre-FC thermocouple is shown; (**e**) The experimental setup, including the compressor, wind tunnel, and programmer load.

**Figure 3 micromachines-12-01432-f003:**
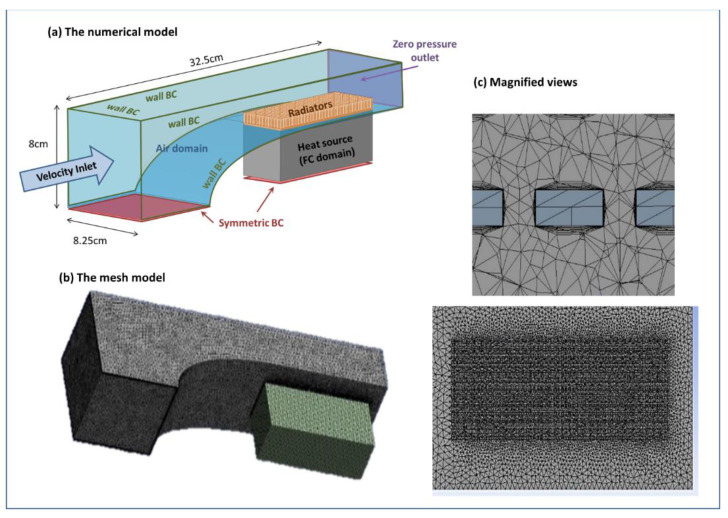
CFD model geometry and mesh. (**a**) The geometry for simulation, representing half of the actual domain, including the tunnel’s ambient air, the FC, and the radiators. Boundary conditions (BC) are specified on the surfaces (**b**) The chosen numerical mesh with 1,846,671 finite volumes and (**c**) magnified views on the mesh of the radiator and the fins.

**Figure 4 micromachines-12-01432-f004:**
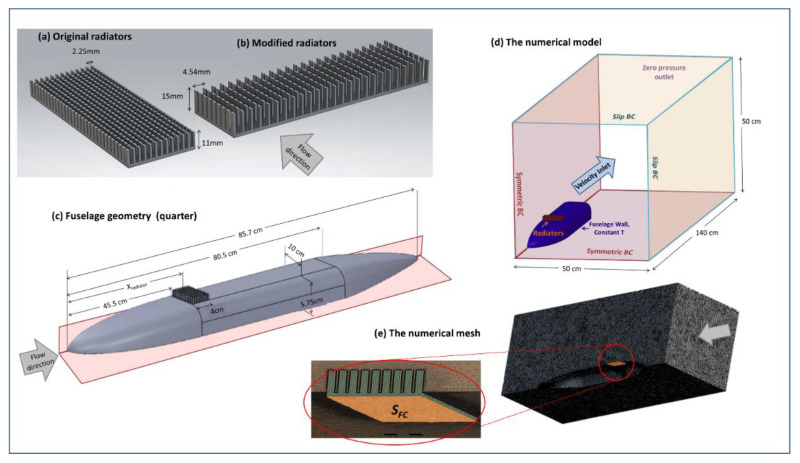
The onboard CFD model. (**a**) The original and (**b**) the modified radiators; (**c**) Geometry of a quarter of the fuselage with radiators. Symmetry boundary conditions are marked in red, radiators location, in respect to fuselage front edge is specified by X_radiator_; (**d**) The geometry of the numerical model, including the domain of ambient air and details of the boundary conditions; (**e**) The numerical mesh, with examples of magnified view around the radiator fins.

**Figure 5 micromachines-12-01432-f005:**
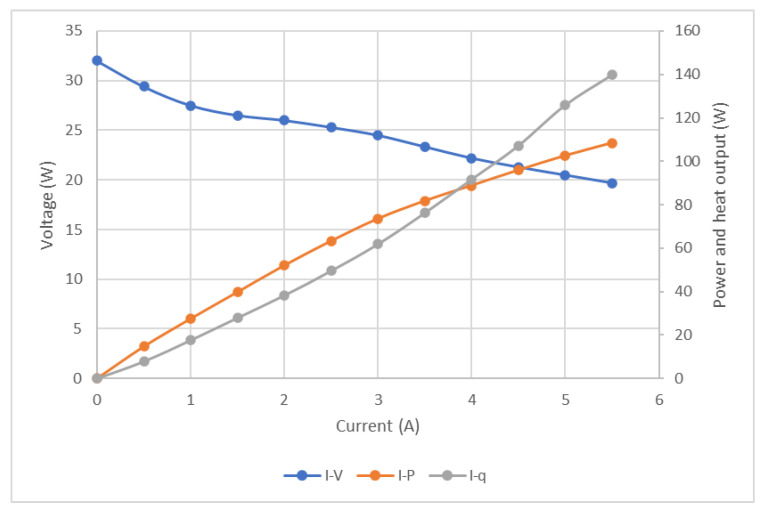
Polarization curve of the utilized FC stack including voltage and power (measured) and heat output (calculated using Equation (2)) as a function of the current.

**Figure 6 micromachines-12-01432-f006:**
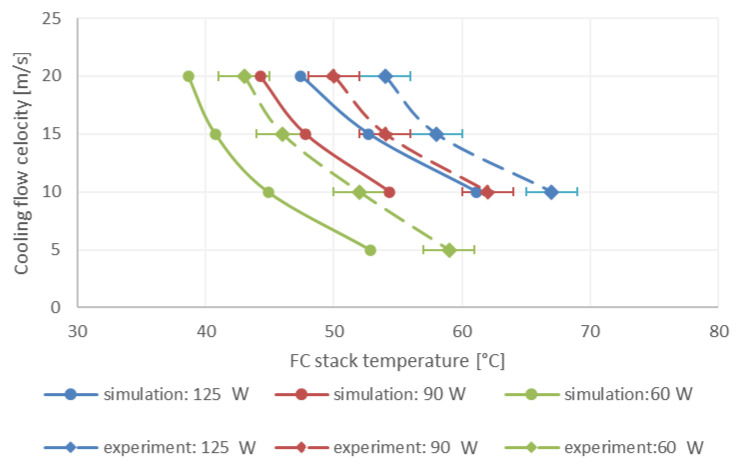
FC maximal temperature as a function of air velocity and consumed power. Comparison between experimental measurements (dashed lines) and simulations results (solid lines).

**Figure 7 micromachines-12-01432-f007:**
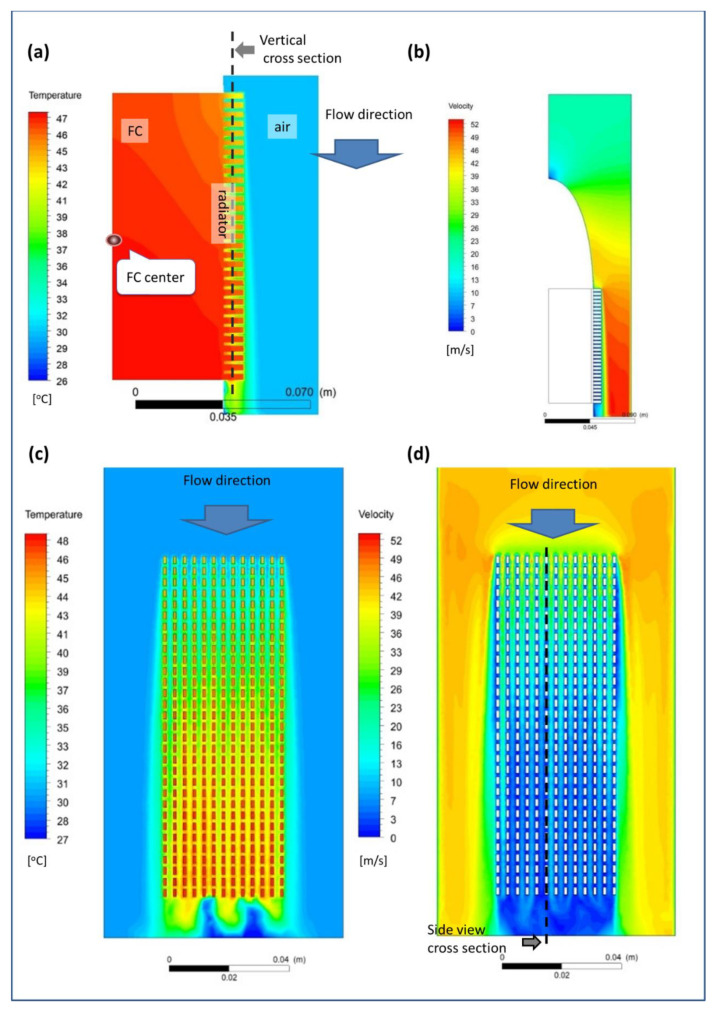
Simulation results of the case with 20 m/s and 125 W heat output. (**a**) Temperature distribution map of the side edge of the FC. FC center is where the temperature is measured; (**b**) Velocity distribution map inside the tunnel (at the cross-section marked in (**d**), located 2 mm distance from the midplane to show also velocities between fins); (**c**) Temperature distribution map at a vertical cross-section (marked in (**a**)) located 1 mm away from the FC; (**d**) Velocity magnitudes at the same surface.

**Figure 8 micromachines-12-01432-f008:**
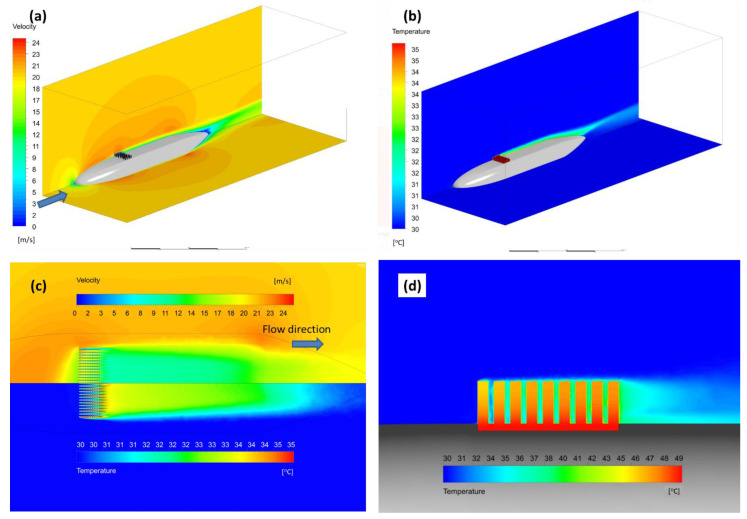
Velocity (**a**) and temperature (**b**) distributions in the model case of 20 m/s at ground atmospheric conditions (30 °C). Magnified top (**c**) and side (**d**) views of the velocity and temperature near and through the radiator fins.

**Figure 9 micromachines-12-01432-f009:**
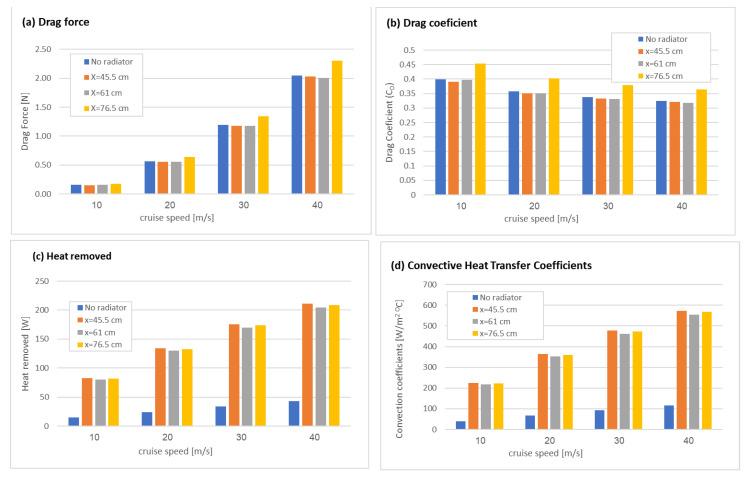
The effect of the radiator location and the cruise speed on drag force (**a**) and heat removed (**c**) for radiators located at X_radiator_ = 45.5 cm, 61 cm, and 76.5 cm from the fuselage front edge and without radiator. Drag coefficient (**b**) was calculated using Equation (4), and heat convection coefficient (**d**) was calculated using Equation (6).

**Figure 10 micromachines-12-01432-f010:**
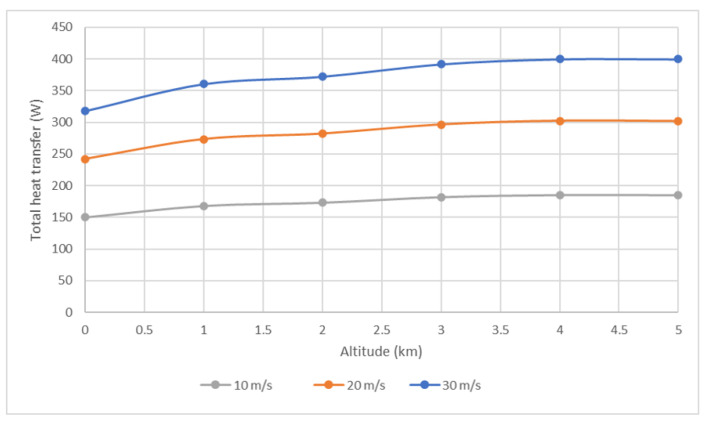
Simulated total heat removal rate at different altitudes and cruise speeds.

**Figure 11 micromachines-12-01432-f011:**
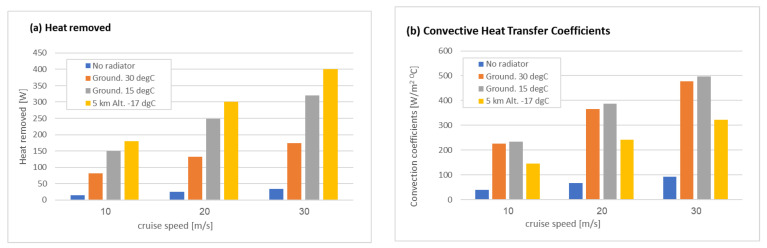
Results summary for the total heat transfer (**a**) and heat transfer coefficient (**b**) for the cases without a radiator, with a radiator on the ground (at 30 °C and 15 °C), and at high altitude (5 km at (−17.5) °C and 47.5 kPa ).

**Table 1 micromachines-12-01432-t001:** Pressure and temperature values and air properties for various drone altitudes.

Altitude (km)	0	1	2	3	4	5
Pressure (Pa)	101,325	90,000	78,567	68,188	58,520	49,586
Temperature (°C)	15	7.5	2.35	−4.5	−11	−17.5
Cp (J/kg K)	1004.1	1003.7	1003.5	1003.3	1003.1	1002.9
Density (kg/m^3^)	1.23	1.12	0.99	0.88	0.78	0.68

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
