# Peer review of "Edge Cooling of a Fuel Cell during Aerial Missions by Ambient Air"

_micromachines, 2021, doi:10.3390/mi12111432_

Round 1

Reviewer 1 Report

Other areas of application of this research would be interesting

Author Response

We thank the reviewer for his/her suggestion. Further non-FC-related applications may include any heat generating device installed on a fixed-wing drone (e.g. battery, engine, CPU, etc.). Following the suggestion, we included this in the discussion section (line 352-353)

Reviewer 2 Report

This work developed a numerical model for the study of fuel cell edge cooling by ambient air in drones. Different parameters including the radiator location, flight speed and altitude were comprehensively investigated in order to optimize the cooling effect. In below please find my comments and questions on it:

1) About the theoretical output voltage of 1.25V, please provide its detailed calculation via the Nernst Equation.

2) About the mesh selection, please provide the grid independence check result.

3) In Table-1, why is the density and Cp not provided?

4) About the model validation, I think Fig. 7 and the associated discussion should be moved before Fig. 6. Also, the 5℃ discrepancy between modeling and experiment may provoke some criticism for the model reliability.

5) Also about Fig. 7, why is there error bar for the simulation result? Shouldn’t it be identical with the same parameter input?

6) Fig. 11 was never discussed in the text.

7) Some typos were found, such as “30 OC”, “15 OC”, “67℃ (for an air velocity of 20 m/s and FC heat output of 60 W)”, “over-colling”, “edge colling”, etc.

Author Response

We thank the reviewer for the comments and suggestions that improved the manuscript. Following the reviewer's suggestions, we added air properties to the table, we explained the difference between the cell potential anode-cathode voltage using the Nernst Equation, further non-FC related applications for the cooling system were suggested, details of the mesh-independence test were added, figures 6 and 7 were switched, and figure 7 was corrected.

see a detailed point-by-point response to the reviewer’s comments in the attached file.

We hope it is now clearer and ready for publication.
